# Review article: A scoping review of human factors in avalanche decision- making

Audun Hetland[1], Rebecca A. Hetland[1], Tarjei T. Skille[1], Andrea Mannberg[1]

[1] CARE (Center for Avalanche Research and Education), UiT - The Arctic University of Norway, Tromsø, 9037, Norway

*Correspondence to*: Audun Hetland (audun.hetland@uit.no)

**Abstract**

The interest in understanding the human aspects of avalanche risk mitigation has steadily grown over the past few decades. Between 2001–2011, 11 research papers on decision-making in avalanche terrain were published in peer-reviewed journals. Between 2012–2022, this number rose to 55. These papers have been authored by researchers from various disciplines and publications in journals across different fields. Despite the field's nascent stage, to guide future research it is pertinent to provide an overview of the insights from existing research literature.

This paper offers a systematic overview of peer-reviewed research on human factors in avalanche decision-making. The overview is based on a systematic literature search covering research published up until the end of 2022. The search was conducted across six databases, including Scopus and Web of Science, using a set of keywords related to avalanche decision-making (e.g., "decision-making," "backcountry skiing," "avalanche terrain," "avalanche accident"). Out of nearly 13,000 articles containing at least one of the key search terms, 70 had a research question related to avalanche decision-making and were published in peer-reviewed academic journals. Additionally, 81 relevant papers were published as ISSW (International Snow Science Workshop) proceedings.

We coded all identified papers based on major and minor research questions, control variables, population covered, and methodology. Twelve concepts described the different research themes (e.g., avalanche accidents, avalanche education, decision-making strategies). Due to a large variation in quality regarding the ISSW papers, we only applied these concepts to the 70 peer-reviewed papers and present them by their main concept. The extracted data from all papers including the ISSW papers can be found at osf.io

## 1 Introduction

### 1.1 Rationale

Approximately 90% of fatal snow avalanche accidents are triggered by the victim or someone in their group (Schweizer and Lütschg, 2001). This underscores that avalanches are more of a human issue than a snow issue.

Over the past two decades, there has been a growing body of research focusing on what has been labelled as 'human factors'. The role of human factors has previously been extensively researched in a range of other scientific fields, e.g., economics, geography, outdoor and recreation, political science, psychology, and public safety and engineering research. It should be noted that the exact definition of the term human factors differs across different disciplines. Within the avalanche research field, human factors have been defined to encompass any human influences that affect the assessment of avalanche risks and the decision-making process (Haegeli et al., 2023). However, even within this literature different research traditions offer different approaches, thus creating a body of knowledge that is heterogeneous in nature. To create a more informative and productive foundation for future research on human factors in avalanche decision-making, we conducted a scoping review.

## 1.2 Objectives

By conducting a scoping review, we wished to examine the extent, range and nature of the evidence so far produced on human factors in avalanche terrain. The following research question has guided this effort:

*What literature exists on how human factors affect decision-making and/or risk assessment done by individuals who expose themselves to avalanche prone terrain?*

The main objectives of our research were:

a. To design and implement a systematic literature search on the topic of human factors in avalanche terrain.

b. To identify relevant literature and extract data from the papers to make a detailed overview of this literature.

## 2 Methods

### 2.1 Scoping review

A scoping review is a type of knowledge synthesis that follows a systematic approach to map evidence on a topic and identify main concepts, theories, sources, and knowledge gaps (Tricco et al., 2018). Unlike systematic reviews, which typically address narrowly focused research questions, scoping reviews cover broader topics and are often used to identify and analyze the extent, range, and nature of research activity in a particular field. By choosing this approach, and by guidance of the PRISMA-ScR checklist, we wished to summarize findings from a body of knowledge that is heterogeneous in both methods and discipline, and to reveal uncharted research areas within the avalanche research field.

### 2.2 Eligibility criteria

Our guiding principle has been that human factors must be central in the included papers. We identified literature where human factors influence actual decision-making or risk assessment while exposed in avalanche terrain, but also in the preparation phase before entering avalanche terrain. Preparation may include both trip planning as well as avalanche education (Greene et al., 2022). Literature focused on decision-making tools was considered relevant in cases where use of the tool is related to human factors in decision-making, but not where the focus is on how the tool relates to weather, terrain, and snowpack aspects. In the following paragraphs we will elaborate and rationalize our criteria for inclusion and exclusion.

### 2.2.1 Publication status

Human factors in avalanche terrain are a nascent research field that has attracted a large interest among both practitioners, stakeholders and users of avalanche terrain. A relatively large share of the literature consists of papers that are not published peer-reviewed (grey literature), mainly as proceedings from the International Snow Science Workshop (ISSW), or as undergraduate and graduate theses (BA, MSc, PhD). The PRISMA guidelines open for including grey literature, and we initially planned to include grey literature. Since they have not gone through a peer-review process we created an additional set of inclusion criteria's where we only included non-peer review papers that 1) contained a clear research question or objective, 2) presented a description of the method used to answer the research question or reach the objective, and 3) built on previous research (i.e., included at least one reference to peer-reviewed research), and 4) did not have a peer-review duplicate. However, our analysis of the

papers revealed a substantial spread in quality even after applying these criteria. While some papers would maybe have been accepted for publication with only minor revisions after a peer-review, others would likely have been given a desk reject. This made it very difficult to develop stringent inclusion criteria. Admittedly, there is also a spread in quality in peer-reviewed articles, but the spread in the grey literature is much larger and since conducting detailed reviews of the quality of the papers is outside of the scope of this paper, we decided to exclude all grey literature. The avalanche research field is different from other research fields, because many practitioners do important research that they present at the ISSW but never even try to publish peer-reviewed. The ISSW conference proceedings are of special importance in this field. We therefore searched through and extracted data from all the 81 ISSW papers that passed the grey literature criteria and organized them thematically in the same way that we did for the peer-reviewed papers. The results can be found at https://osf.io/u9ydm/

### 2.2.2 Participants

All people exposed to avalanche terrain in the backcountry, side country or in out-of-bounds terrain were considered eligible research participants in the included sources of evidence. This includes participants maneuvering avalanche terrain by snow mobiles, snowboard, snowshoes, and skis, and by foot. Recreationalists, professional guides, avalanche safety instructors and educators, ski area patrollers, avalanche professionals (observers, bulletin makers, investigators), as well as other personnel that are expected to personally mitigate and consider avalanche risk (e.g. field geologists, trained soldiers) were included as participants. People appearing as participants through accident reports were also included in the review, as profile information of avalanche victims is considered important information on how human factors may have played a vital role in the decision-making process prior to the avalanche accident. Travelling into avalanche terrain might be self-assisted, snowmobile-assisted, lift-assisted, or motor vehicle-assisted (e.g., helicopter, snowcats).

People travelling by vehicle on roads exposed to avalanche terrain were not included in this review. The rationale behind this is that decisions concerning road risk and safety are made by official authorities, and not by the individuals themselves. Residents living in avalanche exposed areas were excluded from our study by the same rationale.

### 2.2.3 Years considered

In order to include pioneer research and publications that has worked formatively for the development of the field we did not set a lower limit for publication year. Our search has been running up until the end of 2022.

### 2.2.4 Language

Our study has limited its inclusion to sources written in English.

### 2.2.5 Exclusion criteria

We chose to exclude research that focuses strictly on 1) avalanche rescue and medical issues, 2) technical aspects of weather, terrain, avalanche dynamics and forecasting, and 3) management of operations where the decision-maker is not personally affected by the avalanche threat (like risk management in a ski-resort). Our rationale for excluding these important fields is that these research areas do not analyze how individuals personally deal with the threat of being involved in an avalanche accident. We also excluded articles where humans and human behavior in avalanche

terrain is secondary, or implied as part of the research (e.g., extensive accident reports, outdoor or adventure focus). Topics such as decision-making related to rescue after an avalanche has occurred, including medical issues, were not included in the search. Neither were natural science studies or studies primarily focusing on building or technical aspects of avalanche forecasting. However, we note that we did include studies that investigated the effect of avalanche forecast on human factors. Finally, we excluded sources of evidence where the full text was not obtainable, or where human factors were auxiliary or briefly mentioned but were not among the main themes. The excluded topics are also of interest to the scientific community but will require separate searches and are not within the scope of this review.

**2.3. Information sources**

We defined six databases and search engines as relevant to our topic "human factors in avalanche terrain". As the topic is not easily restricted to a specific discipline, *Web of Science* and *SCOPUS* were considered useful sources. They both offer access to multiple databases that reference cross-disciplinary research. Two other discipline specific databases, *PsycINFO* and *Hospitality & Tourism Complete,* were chosen because of the assumption that human factors in avalanche terrain would be published in these academic disciplines. Our previous knowledge of the existing literature led us to this assumption. In addition, we also ran the search in the ISSW proceedings database and *ProQuest* – a database covering dissertations from a range of disciplines. The results from the latter two, primarily originating from the ISSW database, have been subject to the same procedure as the peer-reviewed articles presented in this paper. The results, included the extracted data, can be found in supplementary materials (see https://osf.io/u9ydm/). *Google Scholar* was used as a tool in preliminary searches, and to supplement the final search. We conducted the search between April 27$^{th}$, 2017, and December 31$^{st}$, 2022. Where sources of evidence were found as references or abstracts, but with missing full texts, effort was made to retrieve these texts by requests to relevant libraries or by contacting authors.

**2.4 Search**

**2.4.1 Identifying relevant keywords for systematic search**

We identified keywords using an iterative process. In the first phase, we searched Google Scholar using intuitive search words such as ("human factor in avalanche terrain"). We thereafter used the relevant keywords in the identified papers in a second systematic search: «The Human Factor in Avalanche Terrain".

The keywords and phrases chosen for our search were selected first based on their frequency in the keywords overview (see *keyword, selection.docx* for more details). Other keywords have been added after consulting with researchers familiar with the field. We ran several preliminary searches in the named databases to refine the final set of keywords. The size of the search result has been guiding as to define the relevance and usefulness of the keywords.

**2.4.2 Building the search**

We created two bins, 1) human factor and 2) avalanche. These two bins have a list of associated keywords. Any paper with keywords that matched both bins would be listed as a result. The search is built using the Boolean operators OR and AND, where OR is used between all the keywords within the main categories and AND is used to combine the two categories for the final result. We searched for keywords in titles, abstracts, and listed keywords. Thesaurus terms (pre-defined keywords for specific databases) have been added to the databases with

this functionality. The table below provides an overview of relevant categories of keywords in the two bins (for

more details see *Identifying keywords.docx* and *Keywords, overview.docx* at https://osf.io/u9ydm/).

**Table 1 Overview over keywords included in search**

| Main category "human factor" (combined with OR): | | Main category "avalanche" (combined with OR): |
|---|---|---|
| - Human factor and human error<br>- Decision-making and decision support<br>- Risk (…)<br>- Education and training<br>- Heuristics, cognitive bias and intuition<br>- Situational awareness and pattern recognition<br>- Group dynamics/management/factors<br>- Expertise/expert/professionals and guiding | The two bins are combined with AND.<br><br>Papers with a match in both categories are listed as result | - Avalanche<br>- Backcountry, side-country, off-piste and out of bounds<br>- Skier, snowshoer, snowmobiler, snowboarder<br>- Adventure recreation/tourism |

**2.4.3. Selection of sources of evidence**

The final search result from the individual databases and search engines were added to our library, and duplicates

were filtered out. Guided by our research objectives and eligibility criteria, a preliminary screening was performed

based on title and abstract, separating obviously ineligible studies from possible eligible ones. We used a folder

structure categorizing sources as included, uncertain and excluded. In the next step, two researchers read the full

text. Notes were subsequently compared, and in cases where there was disagreement, the papers were discussed in

depth and a conclusion was drawn based on the extent of how they answered to the research objectives and fulfilled

the eligibility criteria. This process was repeated in three iterations. The final result yielded 70 peer-reviewed

papers. We conducted the same process for the ISSW proceedings.

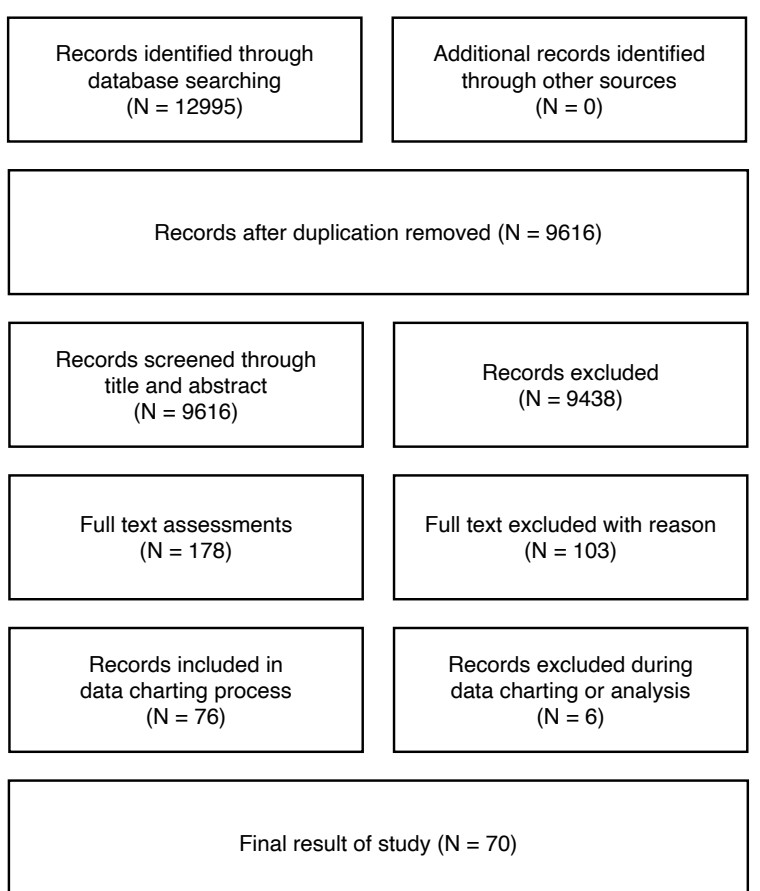

**Figure 1. Flow diagram of the search**

## 2.5. Data charting process

To extract relevant data from the papers, two of the authors developed a matrix schema for charting data from the sources of evidence included. Data was extracted on the basis of year of publication, type of publication, sampling procedure, method of data collection, type of study design, participants (e.g., self- or lift assisted recreationalists, avalanche educators, avalanche forecasters), risk target (the population at risk, e.g., recreationalists, avalanche professionals), focus of study, main explanatory factor, if existing, and, if relevant, control variables of data.

Two independent researchers extracted and coded the data. Notes were subsequently compared and discussed, and if the two coders were not in agreement, or any kind of uncertainty was identifiable, a conclusion was made based on a further discussion with an extended panel of one or two researchers. Table 2 provides a description of the categories of extracted data.

**Table 2 Description of the categories of extracted data from the data charting\***

| Risk target | Population | Sample | Method 1a | Method 2a | Method 3 | Focus 1 + focus 2 | Factor 1 + factor 2 | Control variables |
|---|---|---|---|---|---|---|---|---|
| Recreationalists | Self-assisted recreationalists | Randomized | Survey | Reflection on attitude | Quantitative | DM-errors | FACETS | Socio-demographic |
| General public | Lift-assisted recreationalists | Convenience field | Field observation | Discrete choice experiment | Qualitative | DM-tools | Other heuristic bias | Experience |
| Avy professionals | Heli-assisted recreationalists | Convenience online | Accident analysis | GPS tracks | Mixed design | DM-expertise | Risk perception/attitude | Avy training |
| Avy victims | Motor-assisted recreationalists | Convenience other | Field/lab experiment | User frequency in field | Other | Bayesian perspective | Group dynamics | Avy knowledge |
| Other field workers | Participants of guided groups | Data from sources | Lit. review/overview | Online user frequency | (theoretical, | Risk perception | Other social factors | Avy experience |
| Other | Recreationalists not defined | No sample | Review accidents | Participatory observation | conceptual, | Group dynamics | Leadership | Other variables |
| Tourist industry | Backcountry guides | | Interview | Field experiment | overview etc.) | Demographics | Avy experience | |
| | Ski area patrollers | | Media as data source | Lab experiment | | Avy education | Avy DM competence | |
| | Avy safety instructors/educators | | Review of avy danger | Focus groups/ interviews etc. | | Planning | Avy danger level | |
| | Avy professionals not defined | | Theoretical model | Discourse analysis | | Accidents/incidents | Avy problem | |
| | Avalanche victims (acc. Reports) | | No data collection | Analysis of accidents | | Avy victims | Risk communication | |
| | Professionals field workers | | Critique of theory/tool | Theoretical modelling | | Safety culture | DM-Aid | |
| | Public authorities | | | Comparison to risk in other fields | | Recreation specialization | Goals and policy statements | |
| | Residents in avy exposed terrain | | | Calculated prevention values | | Human factors | Physical activity | |
| | No sample (theoretical etc.) | | | Demographic survey | | Risk communication | Planning / info seeking | |
| | | | | Collection of snow/weather data | | Process of DM | Human factors | |
| | | | | Literature review/ overview | | Safety equipment | Avy education /awareness | |
| | | | | No data collection | | Media/opinions on avy | Recreation specialization | |
| | | | | | | DM related to terrain | Media/opinions of risk | |
| | | | | | | Forecast/danger rating | Weakness in DM-process | |
| | | | | | | | Safety measures/equipment | |

*\*Avy = avalanche (e.g. avy professional – avalanche professional), DM = Decision-making*

### 2.5.1  Categorization of papers according to their main focus

We coded all papers according to their main focus. The different focus themes were developed using an iterative process. One of the authors suggested a first set of themes, based on a previous, non-systematic, review of the literature. During the data's coding process, the two coding researchers could add themes if a paper did not fit the existing themes. In total, 20 themes were identified in the eligible material.

Organizing the literature into 20 themes provides an overview of topics covered in the literature so far. However, some of the topics identified are very narrow, and others overlap. The high number of topics may also make the overview less clear. We therefore decided to revise the codes into a smaller number of research themes. Three of this paper's authors made an initial suggestion of eight research themes. These themes were sent to three international collaborators for feedback and discussion. Based on the discussion, the themes were revised into 12 main research themes (Table 3).

**Table 3. Final research themes.**

| Research theme | Description |
| --- | --- |
| Biases & decision-making errors | All biases and errors |
| Risk communication | Effects of risk communication on learning, understanding, risk perception, decision-making |
| Avalanche education | Effects of avalanche education on learning, and decisions. Content analysis of avalanche education |
| Experience | Experience of travelling in the backcountry and/or assessing avalanche risk. How/what people learn from experience. How experience affects decision-making. |
| Risk perception | Risk judgment, perceived danger/safety. Effects on and of risk perception on decision-making. |
| Willingness to take risk | Measures of risk attitudes. Factors that affect willingness to take risk. Effects of willingness to take risk on decisions. |
| Social factors and group decision-making | Effects of group dynamics and other social factors on individual and group decision making. |
| Avalanche accidents | Factors that affect the risk of being involved in avalanche accidents (incl. accident analysis). Effects of avalanche accidents on decisions, preferences, and perception. |
| Population characteristics | Descriptions of characteristics of certain populations or sub populations. |
| Decision-making strategies | Studies of decision-making tools, strategies, processes, factors. |
| Motivation | Studies on motives for activities and effects of motivation on decision making |
| Methods and theory | Studies that mainly focus on describing/developing new methods or theory |

Two of the authors and the three international collaborators thereafter assigned independently at least one concept to each paper in the dataset. The assignment was based on the focal research question of the article, and not based on the potential relevance for a given research area. For example, studies analyzing avalanche education directly

were assigned the concept 'avalanche education', while studies that might be relevant for avalanche education but did not explicitly investigate the effects of avalanche education or avalanche course curricula were not assigned this concept. Since some papers cover more than one topic, we provided each paper with up to three different concepts. In cases of disagreement, notes were compared and discussed, and concepts were adjusted.

## 3. Result

Of the 12,995 articles that contained at least one of the keywords in the two categories, 76 fulfilled the eligibility criteria and were included in the dataset. During the analysis of the data, we discovered that six of the identified papers did not have human decision-making as their main focus. These papers were therefore removed, and the final data set contained 70 articles.

The eligible papers have publication dates ranging from 1999 to 2022. Over half (N=56) were published in the last 10 years and more than a quarter (N=22) since 2020. Most studies (N=43) rely on quantitative methods. A relatively small number uses qualitative (N=9) or mixed methods (N=11). Only three studies use randomized sampling strategies. Seventy percent rely on convenience samples (N = 50). Sixty-four percent (N = 46) of the articles study backcountry recreationalists. The result from the data charting process with extracted data can be found at https://osf.io/u9ydm/.

### 3.1 Main research themes in the eligible literature

We provide a brief overview of the research themes situated based on research traditions and concepts from related research fields. The list is not meant to cover all potentially relevant research themes on the human dimension of avalanche risk. In Table 5 the papers are sorted on the different research themes.

### 3.1.1 Biases and decision-making errors (N = 11)

A range of cognitive and motivational biases can influence decision making, including those related to risk analysis (Montibeller and von Winterfeldt, 2015), human judgment (Kruglanski and Ajzen, 1983), and strategic planning (Barnes, 1984). The origins of these biases can be traced to both innate and acquired factors, as well as to environmental influences (Croskerry et al., 2013). Despite the prevalence of these biases, individuals often fail to recognize them in their own decision making (Pronin, 2007). Additionally, decision makers can fall into psychological traps such as the anchoring trap and the status quo trap (Hammond et al., 1998).

The papers in this review include a wide range of factors that potentially affect perceptions of risk or skill and/or decisions, like over-confidence (e.g. Bonini et al., 2018), heuristic traps (e.g. Furman et al., 2010), availability affect (e.g. Mannberg et al., 2021a) framing effects (e.g. Stephensen et al., 2021b) but also theoretical (e.g. Zajchowski et al., 2016) and environmental factors (e.g. Wickens et al., 2015). Existing studies in this category typically investigate if people make biased judgements and/or how biases and heuristics affect decision-making in avalanche terrain.

### 3.1.2 Risk communication (N = 9)

Risk communication is a critical aspect of informing the public about potential risks, particularly in public health emergencies (Glik, 2007; Wachinger et al., 2012) and has an impact on risk perception and decision-making (Williams and Noyes, 2007). However, it is often challenging due to the complexity of risk information and the need to consider and understand the audience's beliefs, values and concerns (Fischhoff, 2015; Keeney and

270 von Winterfeldt, 1986). The presentation of risk information can significantly impact its effectiveness, with visual
aids such as graphics playing a key role (Lipkus and Hollands, 1999).

Within the avalanche context, the tag mainly concerns communication via avalanche bulletins. Existing studies in
this category cover both how different groups use and understand the content in avalanche bulletins (e.g.
Fisher et al., 2022) and how the presentation of the information aids or hampers understanding (e.g. Engeset
et al., 2018).

**3.1.3 Avalanche education (N = 4).**
Education plays a crucial role in the ability to conduct risk management in uncertain environments (Carmen Nadia
Ciocoiu and Daniel Neicu, 2007). Education may also help understanding risk and uncertainty (Bob Manson, 2018; Stalker,
2003). The effect of education is pivotal, especially in activities that take place in complex and wicked
environments, where potentially fatal situations are a possibility.

Two of the four existing studies discuss the role of heuristic traps in avalanche courses (Johnson et al., 2020;
Zajchowski et al., 2016). The third study concerns how the processing skills of avalanche bulletin information vary
among recreationists, and how this can be an avenue for continuing education(Fisher et al., 2022). The fourth study
evaluates the effect of avalanche education on risk perception (Greene et al., 2022). It should be mentioned that
many studies use avalanche education as one of many control variables, but these studies are not included under
this tag. The four papers in this category do not cover effects of avalanche education on knowledge and skills, and
analyses of the structure and content of avalanche courses.

**3.1.4 Experience (N = 2)**
Experience can build expertise and therefore significantly impact risk management, but the role of experience in
the risk identification process is much less significant than it is commonly assumed to be (Maytorena et al., 2007).
Particularly, in wicked learning environments where feedback is sparce, experience does not necessarily lead to
expertise (Hogarth et al., 2015).

There are only two papers in this category. One of the studies proposes a new way of measuring expertise (Stewart-
Patterson, 2016). The other investigates how skill affects assessments and understanding of avalanche risk
(Hallandvik et al., 2017). However, several other papers have this as an auxiliary concept, e.g. Landrø (2020)
studies experts' decision-making.

**3.1.5 Risk perception (N = 10)**
Risk perception is a complex phenomenon influenced by various factors and covers both the perceived likelihood
of an outcome, and how dangerous the outcome is perceived to be. Humans have a poor understanding of
probabilities (Hertwig and Erev, 2009). Several studies highlight the role of emotions and cognitive processes in
shaping risk perception (Slovic, 1987; Slovic et al., 2007). Other contributing factors are personal experiences and
cultural factors (Hicks and Brown, 2013; Wachinger et al., 2012) and attitude, risk sensitivity, and specific fear
(Joffe, 2003; Sjöberg, 2000).

In the avalanche literature, studies have focused on a variety of factors that impact risk perception like impact from
experience of fatal avalanche events (e.g. Leiter, 2011), cognitive effect of framing (e.g. Stephensen et al., 2021b),

physical effects of activity (e.g. Raue et al., 2017) or effect of travel strategies (e.g. Michaelsen et al., 2022) or

impact of online user platforms (e.g. Plank, 2016).

**3.1.6 Willingness to take risk (N = 10),**

While risk perception describes a person's understanding of how likely or dangerous a situation is, risk preferences,

or willingness to take risk describe how much they like or dislike the situation given the perceived risk (Dohmen

et al., 2011; Pratt, 1978). Willingness to take risk is tied to demographic factors like gender, age, height, and

parental background (Dohmen et al., 2011), individual factors like sensation seeking (Sharifpour et al., 2013), risk

conception and positive feelings (Dohmen et al., 2018; Isen and Patrick, 1983) or social factors like influence from

peers and mortality salience (Hirschberger et al., 2002; Woodside, 1972) and external factors (Hetschko and

Preuss, 2020; Savage, 1993).

Existing studies in this category typically study how risk preferences correlate with decisions (e.g. Haegeli et al.,

2012; Mannberg et al., 2018), or how willingness to take risk correlate with participant characteristics like gender

and age (e.g. Mannberg et al., 2018; Walker and Latosuo, 2016) or co-hort (e.g. Haegeli et al., 2012; Kopp et al.,

2016) or external factors like equipment (e.g. Haegeli et al., 2014).

**3.1.7 Social factors and group decision-making (N = 6).** Being in a group affects performance and decision

making in multiple ways (Kerr and Tindale, 2004). A group will often outperform individual decision makers

(Kugler et al., 2012; Malone and Bernstein, 2022). However, negative group factors have been repeatedly shown

to decrease decision quality (Kroon et al., 1991) and lead to higher risk taking (Bougheas et al., 2013) and can lead

to fatally flawed decisions (Sunstein and Hastie, 2008). Group size has been shown to be an important predictor,

where large groups can lead to riskier decisions, and challenge communication within groups where groups may

only discuss already shared information and hold back information that is only known to parts of the group (Stasser

and Titus, 1985).

Studies in this category include formation, leadership and decision making in groups (e.g. Zweifel and Haegeli,

2014), social aspiration (e.g. Mannberg et al., 2021b), moral boundaries (Tøstesen and Langseth, 2021), group size

(Zweifel et al., 2016), organizational culture (Johnson et al., 2016) and decision-making within groups, and how

groups affect the decisions made by individuals (e.g. Ebert and Morreau, 2023). There is a large spread in the focus

of existing studies. Topics include group formation, how group size, composition, decision rules affect the quality

of decisions, and how organizational and social norms affect behavior.

**3.1.8 Avalanche accidents (N = 10).**

Accident studies in general offer valuable insights into the causes and prevention of accidents and provide

opportunities for learning (Balasubramanian and Louvar, 2002; Hovden et al., 2011). However, accidents are

complex phenomena which benefit from a comprehensive approach (Cedergren and Petersen, 2011; Moura et al.,

2017). Yet, feedback from experience and accidents are important for improving operational security (Croft, 2020;

Lindberg et al., 2010).

Studies in this category includes trends in accident rates (e.g. Berlin et al., 2019; Page et al., 1999), correlates of

avalanche accidents and demographic factors (e.g. Jekich et al., 2016; Peitzsch et al., 2020), victim profile (e.g.

Soule et al., 2017), group size (e.g. Zweifel et al., 2016), fatality risk in helicopter and snow cat skiing (Walcher et
al., 2019) and organizational culture (Johnson et al., 2016). The existing studies typically characterize avalanche
victims or the situation leading up to the accident.

**3.1.9 Population characteristics (N = 11)**.
People travelling in avalanche terrain are not one homogeneous group, but rather a heterogeneous collection of
people with different motives, skills, ways and means of travel. Tailoring risk mitigation strategies to specific user
groups is crucial for their effectiveness (Bartolucci et al., 2023).

This concept is broad. It includes studies that in some way characterize a "population", regardless of size. Studies
in this category present characteristics for different populations in terms of safety practices (Nichols et al., 2018;
Silverton et al., 2007, 2009), use of avalanche safety equipment (e.g. Ng et al., 2015) and broader focus on human
factor and motivation among different groups (Jackman et al., 2023; Sole et al., 2010).

**3.1.10 Decision making strategies (N = 17)** Decision making under uncertainty is a complex process that requires
a range of strategies (Reale et al., 2023). These strategies can take many forms, from pre-defined (rule-based)
strategies to heuristics (Gigerenzer and Gaissmaier, 2011) or vaguely defined habits (Verplanken and Aarts, 1999).
And in the decision-making process the decision makers need to consider a wide range of potential states and
outcomes, as well as the reliability of information (Hansson, 1996; Polasky et al., 2011). Coping with such
uncertainty requires mental preparedness, agility, and the ability to react to unforeseen events (Kleindorfer, 2008).

The existing literature on decision-making strategies has a very large spread both concerning method and focus.
The studies typically either describe or test relevant strategies, underlying decision-making factors, or use of
decision-making aids in different user groups.
The 17 papers cover both methodological procedures (e.g. Sterchi and Haegeli, 2019; Thumlert and Haegeli, 2017),
as well as empirical collected data on human behavior and mitigation strategies in avalanche terrain (e.g.
Michaelsen et al., 2022). The literature span investigations of professionals (e.g. Løland and Hällgren, 2023) and
recreationists (e.g. Grimsdottir and McClung, 2006), and covers research on decision-making strategies of
backcountry skiers (e.g. Pfeifer, 2009; Witting et al., 2021), mechanized based skiing (e.g. Hendrikx and Johnson,
2016; Sterchi and Haegeli, 2019), as well as snowmobilers (e.g. Baker, 2013; Michaelsen et al., 2022).

**3.1.11 Motivation (N = 3)**
Motivation potentially affects a wide range of factors that drive risk exposure (Kerr and Houge Mackenzie, 2012)
and engaging in analytical thinking (Mækelæ et al., 2023). In the avalanche context, this relates to, e.g., terrain
choices, educational choices, information search, use of risk-mitigation strategies etc.
The concept covers studies that either describe motivational factors in different user groups (Frühauf et al., 2019a),
or how motivations affect decision-making. The three existing papers in this category focus mainly on motives to
seek risk among lift-assisted skiers (Frühauf et al., 2019b, 2020; Fruhaüf et al., 2017).

**3.1.12 Methods and theory (N = 7)**.
The field of social science is characterized by a broad but important variety of theories and methods (Porta and
Keating, 2008). Examples of methods can be observation studies, interviews, surveys and experiments, each with

their own strengths and limitations (Herzog, 1997). It is therefore important to consider the specific research
problem and context when choosing what methodological tools to apply.

The existing studies include papers that develop and describe a new theory or a new empirical method to collect
or analyze data that can help gain a better understanding of human factors in avalanche terrain.
Several of the existing papers in this category present methods for GPS-tracking in combination with surveys, to
collect data on terrain-use and travel behavior in recreational out-of-bounds skiing (Johnson and Hendrikx, 2021;
Sykes et al., 2020). Further, this concept covers methodological investigations to document terrain preferences
(Saly et al., 2020) and terrain selection practices (Thumlert and Haegeli, 2017).

In table 5 the different papers from all the 12 research themes are presented with their different theme tag.

**Table 5.** Eligible papers sorted on main research theme. One paper can be tagged in up to three research themes.

| Author(s) | Title | Year | Tag 1 | Tag 2 | Tag 3 |
|---|---|---|---|---|---|
| Johnson, J; Mannberg, A; Hendrikx, J; Hetland, A & Stephensen, M | Rethinking the heuristic traps paradigm in avalanche education: Past, present and future | 2020 | 1 - Biases & decision-making errors | 3 - Avalanche education | |
| Zajchowski, C. A. B., Browniee, M. T. J., & Furman, N. N. | The Dialectical Utility of Heuristic Processing in Outdoor Adventure Education | 2016 | 1 - Biases & decision-making errors | 3 - Avalanche education | |
| Bonini, N., Pighin S., Rettore, E., Savadori, L., Schena, F., Tonini, S. & Tosi, P. | Overconfident people are more exposed to "black swan" events: a case study of avalanche risk | 2018 | 1 - Biases & decision-making errors | 5 - Risk perception | |
| Stephensen, M. B. & Martiny-Huenger, T. | Liking and perceived safety across judgments of distinct instances of a category of activity | 2021 | 1 - Biases & decision-making errors | 5 - Risk perception | |
| Marengo, D., Monaci, M. G., & Micell, R. | Winter recreationists' self-reported likelihood of skiing backcountry slopes: Investigating the role of situational factors, personal experiences with avalanches and sensation-seeking | 2017 | 1 - Biases & decision-making errors | 6 - Willingness to take risk | |
| Furman, N., Shooter, W., & Schumann, S. | The Roles of Heuristics, Avalanche Forecast, and Risk Propensity in the Decision Making of Backcountry Skiers | 2010 | 1 - Biases & decision-making errors | 2 - Risk communication | 6 - Willingness to take risk |
| Ebert, P. A. | Bayesian reasoning in avalanche terrain: a theoretical investigation | 2019 | 1 - Biases & decision-making errors | | |
| Mannberg, A., Hendrikx, J., Johnson, J. & Hetland, A. | Powder Fever and its Impact on Decision-Making in Avalanche Terrain | 2021 | 1 - Biases & decision-making errors | | |
| Wickens, C. D., Keller, J. W. & Shaw, C. | Human Factors in High-Altitude Mounaineering | 2015 | 1 - Biases & decision-making errors | | |
| Fisher, K., Haegeli, P. & Mair, P. | Exploring the avalanche bulletin as an avenue for continuing education by including learning interventions | 2022 | 2 - Risk communication | 3 - Avalanche education | |
| Terum, J.A., Mannberg, A. & Hovem, F. K. | Trend effects on perceived avalanche hazard | 2022 | 2 - Risk communication | 5 - Risk perception | |
| Haegeli, P., & Strong-Cvetich, L. R. | Using discrete choice experiments to examine the stepwise nature of avalanche risk management decisions-An example from mountain snowmobiling | 2018 | 2 - Risk communication | 6 - Willingness to take risk | 1 - Biases & decision-making errors |
| Clair, A. St., Finn, H., Haegeli, P. | Where the rubber of the RISP model meets the road: Contextualizing risk information seeking and processing with an avalanche bulletin user typology | 2021 | 2 - Risk communication | | |
| Engeset, R. V., Pfuhl, G., Landrø, M., Mannberg, A. & Hetland, A. | Communicating public avalanche warnings - what works? | 2018 | 2 - Risk communication | | |
| Fisher, K., Haegeli, P. & Mair, P. | Impact of information presentation on interpretability of spatial hazard information: lessons from a study in avalanche safety | 2021 | 2 - Risk communication | | |

| | Travel and terrain advice statements in public avalanche bulletins: a quantitative analysis of who uses this information, what makes it useful and what can be improved | 2022 | 2 - Risk communication | | |
|---|---|---|---|---|---|
| Fisher, K., Haegeli, P. & Mair, P. | | | | | |

411

| Author(s) | Title | Year | Tag 1 | Tag 2 | Tag 3 |
|---|---|---|---|---|---|
| Greene, K., Hendrikx, J. & Johnson, J. | The Impact of Avalanche Education on Risk Perception, Confidence, and Decision-Making among Backcountry Skiers | 2022 | 3 - Avalanche education | 5 - Risk perception | |
| Landrø, M., Engeset, R. & Pfuhl, G. | The role of avalanche education in assessing and judging avalanche risk factors | 2022 | 3 - Avalanche education | | |
| Hallandvik, L., Andresen, M. S., & Aadland, E. | Decision-making in avalanche terrain-How does assessment of terrain, reading of avalanche forecast and environmental observations differ by skiers' skill level? | 2017 | 4 - Experience | 10 - Decision making strategies | |
| Stewart-Patterson, I. | Measuring decision expertise in commercial ski guiding in a more meaningful way | 2016 | 4 - Experience | 12 - Methods and theory | |
| Stephensen, M. B.; Schulze, C.; Landrø, M.; Hendrikx, J. & Hetland, A. | Should I judge safety or danger? Perceived risk depends on the question frame | 2021 | 5 - Risk perception | 1 - Biases & DM errors | |
| Groves, M. R. & Varley, P. J. | Critical mountaineering decisions: technology, expertise and subjective risk in adventurous leisure | 2020 | 5 - Risk perception | 6 - Willingness to take risk | |
| Plank, A. | The hidden risk in user-generated content: An investigation of ski tourers' revealed risk-taking behavior on an online outdoor sports platform | 2016 | 5 - Risk perception | 2 - Risk communication | |
| Mehus, G., Mehus, A. G., Germeten, S. & Henriksen, N. | Young people and snowmobiling in northern Norway; accidents, injury prevention and safety strategies | 2016 | 5 - Risk perception | | |
| Raue, M., Streicher, B., Lermer, E., & Frey, D. | Being active when judging risks: bodily states interfere with accurate risk analysis | 2017 | 5 - Risk perception | | |
| Leiter, A. M. | The sense of snow - Individuals' perception of fatal avalanche events | 2011 | 5 - Risk perception | | |
| Kopp, M., Wolf, M., Ruedl, G. & Burtscher, M. | Differences in Sensation Seeking Between Alpine Skiers, Snowboarders and Ski Tourers | 2016 | 6 - Willingness to take risk | 9 - Population characteristics | |
| Walker, E., & Latosuo, E. | Gendered decision-making practices in Alaska's dynamic mountain environments? A study of professional mountain guides | 2016 | 6 - Willingness to take risk | 9 - Population characteristics | |
| Haegeli P., Gunn M., & Haider W. | Identifying a High-Risk Cohort in a Complex and Dynamic Risk Environment: Out-of-bounds Skiing-An Example from Avalanche Safety | 2012 | 6 - Willingness to take risk | 12 - Methods and theory | |
| Haegeli, P., Rupf, R. & Karlen, B. | Do avalanche airbags lead to riskier choices among backcountry and out-of-bounds skiers? | 2020 | 6 - Willingness to take risk | | |
| Mannberg, A., Hendrikx, J., Landrø, M., & Ahrland Stefan, M. | Who's at risk in the backcountry? Effects of individual characteristics on hypothetical terrain choices | 2018 | 6 - Willingness to take risk | | |

| Author(s) | Title | Year | Tag 1 | Tag 2 | Tag 3 |
|---|---|---|---|---|---|
| Johnson, J., Haegeli, P., Hendrikx, J., & Savage, S. | Accident causes and organizational culture among avalanche professionals | 2015 | 7 - Social factors and group decision making | 8 - Avalanche accidents | |
| Zweifel, B., Procter, E., Techel, F., Strapazzon, G., & Boutellier, R. | Risk of Avalanche Involvement in Winter Backcountry Recreation: The Advantage of Small Groups | 2016 | 7 - Social factors and group decision making | 8 - Avalanche accidents | |
| Mannberg, A., Hendrikx, J. & Johnson, J. | Risky positioning – social aspirations and risk-taking behaviour in avalanche terrain | 2020 | 7 - Social factors and group decision making | | |
| Ebert, P. A. & Morreau, M | Safety in numbers: how social choice theory can inform avalanche risk management | 2022 | 7 - Social factors and group decision making | | |
| Tøstesen, G & Langseth, T | Freeride skiing - Risk-taking, Recognition, and Moral Boundaries | 2021 | 7 - Social factors and group decision making | | |
| Zweifel, B., & Haegeli, P. | A qualitative analysis of group formation, leadership and decision making in recreation groups traveling in avalanche terrain | 2014 | 7 - Social factors and group decision making | | |
| Berlin, C., Techel, F., Moor, B. K., Zwahlen, M., Hasler, R. M. & Swiss Natl Cohort Study, Grp | Snow avalanche deaths in Switzerland from 1995 to 2014-Results of a nation-wide linkage study | 2019 | 8 - Avalanche accidents | 9 - Population characteristics | |
| Soule, B., Reynier, V., Lefevre, B., & Boutroy, E | Who is at risk in the French mountains? Profiles of the accident victims in outdoor sports and mountain recreation | 2017 | 8 - Avalanche accidents | 9 - Population characteristics | |
| Techel, F., Zweifel, B., & Winkier, K. | Analysis of avalanche risk factors in backcountry terrain based on usage frequency and accident data in Switzerland | 2015 | 8 - Avalanche accidents | | |
| Jekich, B. M., Drake, B. D., Nacht, J. Y., Nichols, A., Ginde, A. A. & Davis, C. B. | Avalanche Fatalities in the United States: A Change in Demographics | 2016 | 8 - Avalanche accidents | | |
| Page, C. E., Atkins, D., Shockley, L.W. & Yaron, M. | Avalanche deaths in the United States: a 45-year analysis | 1999 | 8 - Avalanche accidents | | |
| Peitzsch, E.; Boilen, S.; Logan, S.; Birkeland, K. & Greene, E. | Research note: How old are the people who die in avalanches? A look into the ages of avalanche victims in the United States (1950–2018) | 2020 | 8 - Avalanche accidents | | |
| Walcher, M.; Haegeli, P. & Fuchs, S. | Risk of death and major injury from natural hazards in Helicopter and Snowcat skiing in Canada | 2019 | 8 - Avalanche accidents | | |
| Nichols, T. B., Hawley, A. C., Smith, W. R., Wheeler III, A. R., & McIntosh, S. E. | Avalanche Safety Practices Among Backcountry Skiers and Snowboarders in Jackson Hole in 2016 | 2018 | 9 - Population characteristics | 10 - Decision making strategies | |
| Ng, P., Smith, W. R., Wheeler, A., & MacIntosh, S. E. | Advanced Avalanche Safety Equipment of Backcountry Users: Current Trends and Perceptions | 2015 | 9 - Population characteristics | | |
| Sole, A. E., Emery, C. A., Hagel, B. E., & Morrongiello, B. A. | Risk Taking in Avalanche Terrain: A Study of the Human Factor Contribution | 2010 | 9 - Population characteristics | | |
| Jackman, P. C., Hawkins, R. M., Burke, S. M., Swann, C. & Crust, L. | The psychology of mountaineering: a systematic review | 2020 | 9 - Population characteristics | | |

| Silverton, N. A., MacIntosh, S. E., & Kim, H. S. | Avalanche safety practices in Utah | 2007 | 9 - Population characteristics | | |

415

| Silverton, N. A., McIntosh, S. E., & Kim, H. S. | Risk Assessment in Winter Backcountry Travel | 2009 | 9 - Population characteristics | | |

| Author(s) | Title | Year | Tag 1 | Tag 2 | Tag 3 |
|---|---|---|---|---|---|
| Grimsdottir, H., & McClung, D. | Avalanche risk during backcountry skiing - An analysis of risk factors | 2006 | 10 - Decision making strategies | 8 - Avalanche accidents | |
| Michaelsen, B., Stewart-Patterson, I., Rolland, C. G., Hetland, A. & Engeset, R. V. | Behaviour in Avalanche Terrain: An Exploratory Study of Illegal Snowmobiling in Norway | 2022 | 10 - Decision making strategies | 9 - Population characteristics | |
| Sterchi, R. & Haegeli, P. | A method of deriving operation-specific ski run classes for avalanche risk management decisions in mechanized skiing | 2019 | 10 - Decision making strategies | 12 - Methods and theory | |
| Thumlert, S. & Haegeli, P. | Describing the severity of avalanche terrain numerically using the observed terrain selection practices of professional guides | 2017 | 10 - Decision making strategies | 12 - Methods and theory | |
| Baker, J., & McGee, T. K. | Backcountry Snowmobilers' Avalanche-Related Information-Seeking and Preparedness Behaviors | 2016 | 10 - Decision making strategies | | |
| Hendrikx, J., Johnson, J., & Shelly, C. | Using GPS tracking to explore terrain preferences of heli-ski guides | 2016 | 10 - Decision making strategies | | |
| Landro, M.; Hetland, A.; Engeset, R. V. & Pfuhl G. | Avalanche decision-making frameworks: Factors and methods used by experts | 2020 | 10 - Decision making strategies | | |
| Løland, S. & Hällgren, M. | ´Where to ski?´: an ethnography of how guides make sense while planning | 2022 | 10 - Decision making strategies | | |
| Sterchi, R., Haegeli, P. & Mair, P. | Exploring the relationship between avalanche hazard and run list terrain choices at a helicopter skiing operation | 2019 | 10 - Decision making strategies | | |
| Witting, M., Filimon, S. & Kevork, S. | Carry along or not? Decision-making on carrying standard avalanche safety gear among ski tourers in a German touring region | 2021 | 10 - Decision making strategies | | |
| Haegeli, P, Haider, W, Longland, M, & Beardmore | Amateur decision-making in avalanche terrain with and without a decision aid: a stated choice survey | 2010 | 10 - Decision making strategies | | |
| Landro, M., Pfuhl, G., Engeset, R., Jackson, M. & Hetland, A. | Avalanche decision-making frameworks: Classification and description of underlying factors | 2020 | 10 - Decision making strategies | | |
| McCammon, I., & Haegeli, P. | An evaluation of rule-based decision tools for travel in avalanche terrain | 2007 | 10 - Decision making strategies | | |
| Pfleifer, C. | On probabilities of avalanches triggered by alpine skiers. An empirically driven decision strategy for backcountry skiers based on these probabilities | 2009 | 10 - Decision making strategies | | |

418

| Author(s) | Title | Year | Tag 1 | Tag 2 | Tag 3 |
|---|---|---|---|---|---|
| Fruhauf, A., Anewanter, P., Hagenauer, J., Marterer, N. & Kopp, M. | Freeriding-Only a need for thrill? Comparing different motives and behavioural aspects between slope skiers and freeride skiers | 2019 | 11 - Motivation | 6 - Willingness to take risk | |
| Fruhauf, A., Hardy, W., Pfoestl, D., Hoellen, F. G. & Kopp, M. | A qualitative approach on motives and aspects of risk in freeriding | 2017 | 11 - Motivation | | |
| Fruhauf, A., Zenzmaier, J. & Kopp, M. | Does Age Matter? A Qualitative Comparison of Motives and Aspects of Risk in Adolescent and Adult Freeriders | 2020 | 11 - Motivation | | |
| Sykes, J.; Hendrikx, J.; Johnson, J. & Birkeland, K. W. | Combining GPS tracking and survey data to better understand travel behavior of out-of-bounds skiers | 2020 | 12 - Methods and theory | 10 Decision making strategies | |
| Johnson, J & Hendrikx, J. | Using Citizen Science to Document Terrain Use and Decision-Making of Backcountry Users | 2021 | 12 - Methods and theory | | |
| Saly, D.; Hendrikx, J.; Birkeland, K. W.; Challender, S. & Johnson, J. | Using time lapse photography to document terrain preferences of backcountry skiers | 2020 | 12 - Methods and theory | | |

419
420

## 4. Discussion

Our review shows that the number of peer-reviewed papers on the human factors in avalanche decision-making has increased substantially during the past decade. The vast majority of published studies use convenience sample methods to collect, and quantitative methods to analyze data from their participants, which mainly consists of recreational backcountry users (especially skiers). In this study we only include papers describing how human factors influence actual decision-making or risk assessment for those exposed to avalanche risk. However, there is a number of related topics that also should be explored, like avalanche rescue and medical issues, technology or solutions to assist decisions or mitigate avalanche risk including avalanche forecasting, management and decision making in operations where the decision maker is not personally affected, and many others.

Our review of research themes suggests that most papers have research questions related to 'biases and decision-making errors' (N =11), 'risk communication' (N = 9), 'risk perception' (N = 10) or 'willingness to take risk' (N = 10). Many of the papers provide descriptions of the behaviors or characteristics of specific groups of backcountry users. These papers were often categorized as 'population characteristics' (N = 11) or 'decision-making strategies' (N = 17). However, we would like to highlight that, given the large variety of studies included, the two latter research themes are broader and thus less informative than the other themes.

Within each category there are gaps and interesting questions for future studies. The studies within each category could have been explored in more detail, for example, through narrative reviews, and compared to studies beyond the avalanche literature through gap analysis. This is beyond the scope of this study but a worthwhile effort for future studies. We do however note that the literature on important topics like social factors (N= 6), motivation (N=3), experience (N=2) and avalanche education (N=4) is very limited, and therefore not suitable for narrative reviews. We therefore would like to tie some comments on why these are important, potential questions to ask and some reflections on how to approach them.

### 4.1. Social factors and group decision making

Most decisions are made by groups, not individuals. This is especially the case for recreational decision-making in avalanche terrain. The sociality of humans further means that our decisions are very susceptible to the influence of people around us and this affects decision making in multiple ways (Kerr and Tindale, 2004).
At its best, groups can easily outsmart individuals (Malone and Bernstein, 2022). However, individuals within groups are subject to a number of dynamics that influence decision-making beyond their immediate control. These dynamics can lead them into pitfalls and dilemmas that could potentially be mitigated with greater knowledge and awareness of typical social mechanisms present in groups navigating avalanche terrain.. At its worst, groups can have detrimental or even catastrophic effect on decision-making (Cartwright, 1973; Janis, 2008). Determining factors include group size and composition, formation and leadership, communication and skill, social aspiration, culture and moral, cohesion and trust. Only a few of these topics have received attention in avalanche literature and many important questions remain unexplored. .

## 4.2. Motivation

Motivation affects a wide range of behaviors that can propel people to search for information or use products and services designed to improve their decisions. However, people have different motives for the same activity (Hornby et al., 2024). This variability suggests that motivation is not only a driver of behavior but also a potential source of bias, especially when strong motivation leads to an overshadowing or underestimation of cumulative risks, as observed in contexts involving appealing or high-stakes outcomes (Knäuper et al., 2005). Such motivational biases can result in individuals disregarding potential risks or rationalizing behavior that may compromise long-term well-being. In this study we only found three papers that specifically focus on motivation, and even here the focus is more toward slope and freeriding. An investigation of motives for different segments of backcountry skiers, maybe separating between genders, terrain choices or locals vs tourists is warranted. A systematic review study on motivation in extreme sport (Hornby et al., 2024) found that the more self-efficacy people had in their activity the more risk they we willing to take. However, unlike many other sports, the major hazards of avalanches are not directly tied to mastering skiing, and the dynamics of self-efficacy in particular, or in motivation more generally may be different than in other risk prone activities.

## 4.3. Experience

In an environment with high quality feedback, experience may translate to expertise (Ericsson, 2008). This is unfortunately not the case in avalanche terrain. The inherent lack of feedback creates a wicked learning environment (Hogarth et al., 2015). In addition, avalanche assessments are complex, even for trained experts (Landrø et al., 2020) and without a first-hand experience of avalanche accidents the risk is abstract (Hetland et al., 2024) leaving fear to be among the least prominent emotions among skiers (Hetland et al., 2018).

As in many other fields, the absence of catastrophic events often presents a unique challenge for accurately assessing risk and guiding future actions. While an avalanche provides clear feedback that informs risk perception and promotes preventative measures, the lack of such an event can lead to cognitive biases and distorted risk assessments. This phenomenon, sometimes described as "the dog that didn't bark," occurs when individuals or societies overlook potential risks because they have not recently experienced adverse events (Kahneman and Tversky, 2013). The role of experience is therefore important in order to understand how the absence of avalanches events can lead to complacency, overconfidence, and behavior based on perceived, rather than actual, risk levels (Stephensen et al., 2021a). The two studies presented in this review provide a first take on how to assess expertise decoupled from experience (Stewart-Patterson, 2016) and the role of experience and behavioral consequences across skill level (Hallandvik et al., 2017). Understanding how decision-makers interpret—or ignore—the absence of negative feedback is essential for developing frameworks that ensure effective education or risk management or promote sustainable behaviors in the face of low-probability, high-impact events like avalanches.

## 4.4. Avalanche education

Avalanche education provided by trained instructors ideally leads to improved skills in risk assessment and mitigation. However, we have not found any papers analyzing the quality of avalanche education, or how courses can be improved to increase learning. The studies in this review underscore that decision-making in avalanche terrain is a complex process with many moving parts in uncertain environments where feedback is fickle.  However, when people are most often the cause and victims of injury and death in avalanche terrain, the crux of the problem is avalanche education. How do people come to understand and later manage those complex factors? To date, avalanche education research sorely lacks careful studies of how people are taught and learn relevant knowledge and skills, and how people keep their knowledge and skills current. What knowledge and skills are essential and when? Which ways of learning are most effective, and how do they work? How is effective avalanche education made readily available to those who need it, and how do we assure that they get it for not only their own safety, but the safety of others? How does avalanche education change behavior? And does avalanche education leave people less exposed to risk or does it in fact make people more susceptible to expose themselves to a risk they may not fully appreciate? (Yudkowsky, 2008). These questions deserve urgent, interdisciplinary research attention.

## 4.5. Methodological approaches

Most of the papers included in this scoping review rely on a quantitative analysis of cross-sectional convenience samples, i.e., participants are recruited via personal networks, social media, or via avalanche organizations, and are only observed once. Most studies extract information via surveys. While these kinds of analyses can increase our understanding of some factors that affect decisions in avalanche terrain, the conclusions that can be drawn from the analyses are limited. There are several reasons for this.

Using convenience sampling via 'avalanche networks' means that the researcher is more likely to reach participants with some form of interest in avalanche safety (e.g., visiting the avalanche bulletin website). In addition, among the individuals reached, those with a greater interest in avalanche safety are more likely to complete their participation. Since both learning and decision-making likely depend on interest, results from studies relying on connivence samples may not hold for the general population at risk for avalanches.

Non-experimental cross-sectional analysis can identify *correlations* between different factors (e.g., avalanche education/avalanche bulletin use and avalanche accidents), but cannot identify *causal* mechanisms or the *direction* of causation. There are several reasons for this, one of which is self-selection. Like with participation in research studies, participation in avalanche courses and reading the avalanche bulletin likely covary with the interest to venture into avalanche terrain (or with avalanche safety). In other words, finding that avalanche training/reading the bulletin correlates with experience of avalanche incidents or terrain choices is not sufficient to draw the conclusion that courses or forecasts have a *causal* effect on risk exposure. Experimental studies randomly assign participants to different 'treatments' (participating in a course, reading the bulletin). As such, these studies avoid the selection problems described above. Non-experimental longitudinal studies (studies that follow people over time) have issues with self-selection but can evaluate *changes* in behavior and preferences before and after an event.

This makes it possible to identify causal effects on a specific group of participants, even if it is not possible to generalize the results to the general population.

Finally, surveys that ask participants about their stated preferences and experiences can elicit information about what people think that they would feel and do in different situations, or what they remember from past situations. However, people in general are poor at predicting how they will feel and act in situations that are different from their current one (Mathews and Bradle, 1983; Thomas and Diener, 1990). In addition, humans' need to preserve a positive self-image can affect how we remember and explain past experiences (e.g. Alicke and Sedikides, 2009). In situ studies, that observe participants in the field when the experiences occur, therefore hold potential to reveal mechanisms that surveys fail to find.

## 4.6. Limitations

The spreadsheet containing the data from eligible papers has some limitations that should be kept in mind when used. First, to systematically assign a main concept to a paper, we focused on the paper's primary objective and focal research question. However, human factors in avalanche decision-making are a complex concept, and a single paper can encompass insights relevant to a multitude of topics. In addition, while all included studies are published peer-reviewed, the clarity of the research question, and the link between the research question and analysis, vary substantially in the final dataset. The resulting concepts may therefore provide an overly simplistic picture of the content in the current literature. Much of the literature offers insights that extend to topics beyond their main concept, and the resulting categorization should not be considered a measure of topic inclusion.

Second, while the data extraction and organization of the material followed a structured procedure, the evaluation was done by a limited number of researchers. This means that the papers have been interpreted through the lens of a few individuals. The evaluation is therefore subjective, and other researchers may have categorized the data differently.

Finally, the methodological decisions relating to the eligibility criteria, publication status, years and languages considered, and information sources for the literature were aimed to create a more systematic review. While these decisions improved the relevance, consistency, and quality of the studies, they have drawbacks in that they inherently create a publication bias. As a result, the current study is biased towards Western academic perspectives in predominantly European and North American industry contexts. However, given that this study is a first attempt to consolidate this body of research from across the widely dispersed and inconsistent publishing outlets utilized by the avalanche community, it serves as a fundamental first step toward building subsequently more comprehensive and inclusive overviews of the literature.

## 5. Conclusion

The aim of the systematic literature search was to provide an overview of the existing body of research on human factors in avalanche decision-making. We hope the shared spreadsheet and the organization of the literature into different research themes will help researchers find relevant literature and identify important knowledge gaps that remain to be filled.

We would like to end with a call for action. The work with this literature search has been challenging for mainly two reasons. First, many papers lack clear and relevant keywords. This made it difficult to identify them in our search. Second, some of the papers proved difficult to access, even after trying to contact authors or libraries. We would therefore envision a shared database similar to *PsychInfo* with categorization of studies in various categories and we encourage authors to publish their papers open access so that important messages are not locked in behind pay walls. This is particularly important given that the readership may be practitioners without access to scientific libraries. Finally, we encourage researchers within the field to draw attention to existing gaps that should be closed, where assessing the quality of avalanche education is most compelling.

## 6. Author contribution

AH lead the project and has been involved in all stages of the project including design, implementation, and writing and editing paper. RAH: designed and ran the search, developed the sorting procedure, writing and editing, TTS: Finalizing sorting, writing and editing, AM: advice of design and implementation, writing and editing.

## 7. Acknowledgment

We would like to thank everyone that has contributed to the sorting process, particularly Finn Hovem, Ingrid Stette Haaberg and Markus Aase for their extended and laborious effort in the initial screening. We would also like to thank our three international collaborators Pascal Haegeli, Ann St. Clair and Kelly McNeil for valuable help in identifying the concepts and contributing to the proceedings paper version of this manuscript that was presented at ISSW in Bend October 2023.

### 7.1 Funding

This study was partly funded by NordForsk grant 105061.

## 8. Conflict of interest

The authors declare they have no conflict of interest.

## 9. Data availability

All relevant data for this study can be downloaded at https://osf.io/u9ydm/

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
