# Peer review of "Review article: A scoping review of human factors in avalanche decision- making"

_EGUsphere, 2024_

## Author Response (AR2)

**Answer to reviewers – Literature review human factor**

REVIEWER 1
**Comment**: Thank you for the opportunity to review this paper. The authors provide a very valuable overview of the existing scientific literature and its content on the topic of human factors in avalanche decision making. I fully agree with the authors that although much research has been done on this topic in the last decade, it still has a lot of potential.

The present study is very carefully scientifically structured and excellently written in this paper. I congratulate the authors! Even after studying this paper in detail, I have not discovered any serious shortcomings and **therefore recommend the editor to publish this paper**. I have the following **minor comments**:

**Answer**: Dear dr.Zweifel.

Thank you very much for taking the time to review our paper and also for the detailed comments where you point out lack of clarity or mishaps that we our selves were unable to pick up in the final stage of writing up the paper.

We will address them in turn:

**Comment:** There is confusion on the number of papers in the results: in the abstract, on page 5 and page 9 (line 285) you mention 70 papers, in Fig. 1 you state 69 papers and in the Table 2 I counted 69 papers?

**Answer:** Thank you very much for pointing out this. In the final stage of writing up the paper we went back and checked the final analysis one last time. We then discovered that one of the papers had been falsely discarded  and therefore included this in the papers, which increased the number of papers from 69 to 70. We have apparently not been able to implement this change through out the paper and we have corrected this througout the paper and figure. We have also added the final paper to the table 5.

**Comment:** Page1, line 40: I was a bit confused by the term 'qualitative systematic scoping review' since you later distinguish *scoping review* vs. *systematic review* (paragraph 2.1). Accordingly, I suggest skipping 'systematic' here.

**Answer:** Thank you for pointing out the lack of coherence in our terminology. We have changed this to scoping review

**Comment:**Page3, line 105-106: It makes totally sense to me to not set a lower limit for publication year. One argument could also be, that this topic had been under research only in recent times anyway.

**Answer:** Thank you. Yes we where curious to see what had been written about human factor back in time and therefore searched without any lower limit on publication year. However, when reviewing the results we found that all papers that matched our criterias where of newer date. So even though we did not have any lower limit we did not find any substantial contributions dating back more than two decades.

**Comment:**Page 5, Table 1: I'm not familiar with the term 'off-bounds'. I know 'off-piste' or 'out of bounds'...?
**Answer:** Thank you for pointing this out. This is mean to be out-of-bounds and we have correct this in the table.

**Comment:** Page 9, paragraph 3.1.1: I was very surprised to not find Ian McCammon in this paragraph, since he really was the pioneer within the heuristic traps field. I guess, this is, because he didn't publish his findings in peer-reviewed journals? Probably this can be discussed (discussion or limitation section)?

**Answer:**Yes, we fully agree with you that McCammon should be given a attention and credit as his ISSW proceedings are hallmark papers in the human factor research. We initially aimed to include conference proceedings in our search. Indeed we have conducted the same search in the Montana library where the ISSW proceedings are stored - as well as other databases where conference proceedings are listed. We found 80 relevant proceeding papers solely originating from the ISSW conference. Many of these are important studies for the avalanche community and we therefore conducted the same data extraction and listing of these papers which we provide at https://osf.io/u9ydm/ However, the spread in quality was substantial even after we introduced a nother set of selection criterias. We therefore finaly decided to only incldude peer-reviewed papers in this review. We have motivated the decision in the paper and also mentioned McCammons work spesifically.

**Comment:** Page 14, Table 5: I suggest using exactly the same wording for the tags (Tag 1 to Tag 3) as used in Table 3. Then it seems that there is some formatting issues: I assume that there are different text sizes in this table and there is an unpleasant table break form page 16 to page 17
**Answer:** Thank you. We have made sure it is the same font (Arial regula - 8) and also aligned the tags name from table 3 to table 5.

REVIEWER 2
General comments
1. I greatly appreciate the opportunity to review this wonderful paper. The authors have embarked upon an ambitious and pioneering review that establishes an important resource for follow-on work. Table 5 in particular will be invaluable for future researchers in this field. Overall, the work is well conceived, well executed and very readable. I concur with the previous reviewer that this paper is worthy of publication with some very minor revisions.

**Answer:** Dear reviewer, thank you for using both time and effort to review our manuscript. We are pleased to read that you found that the paper makes an important contribution, and we highly appreciate your insightful and constructive comments. On our side we have been busy with organizing the ISSW in Tromsø, and apologize for our rather late response. Below, we describe how we will try to accommodate your suggestions and concerns.
Specific comments

1. First, I agree with the excellent points raised by the previous reviewer. In particular, I appreciate the author's willingness to address the use of the term "systematic": I too found the term confusing with regard to the intent of the research effort. Simple wording changes would fix this issue.

**Answer:** We agree with you and the other reviewer that the description "Qualitative systematic scooping review" is confusing. A scoping review is a systematic review, so the term qualitative is confusing. We have deleted "qualitative and systematic" so that the description now reads "scoping review".

1. Page 1, lines 35-6: As I am sure the authors are aware, the term "human factors" as used in the avalanche community differs appreciably from its meaning in the broader scientific and engineering literature. A sentence here about this usage discrepancy will avoid confusing readers from other fields.

**Answer:** We agree that the term 'human factors' should be defined and has added a description of how human factor including how we define the term.

1. Page 2, lines 57, 61-2: The authors define one of the functions of a scoping review as identifying knowledge gaps. They also state that one of their goals was to reveal "uncharted research areas" within the field. Section 4 could be improved with a more explicit discussion of these gaps and unexplored research areas. This brief addition would be a valuable take-away for many readers.

**Answer:** We completely agree that identifying gaps in the literature is important. We have added an overarching description of this in section 4 – and also a more indepth description of the categories where we find very few papers (social factors, experience, motivation and avalanche education). Here we point to potential venues for reseach. We end section 4 with a discussion on different methodological approached and considerations. This has substantially expanded the discussion – and also increased the length of the paper. However, even though the paper is long we think it makes sence to add this to the discussion.
For the other categories where there is larger streams of research we belive they would need more attention to reveal and discuss gaps and potential future venues for research. This is something that we point out in the discussion and encourage for future research. We hope this is a good take away for the reader – even though we do not point to gaps within all categories.

1. Page 3, Section 2.2.1: I found the treatment of ISSW papers confusing. The link leads to a coding of 81 papers, suggesting by their sheer number that these are important for a comprehensive scoping review. These papers are mentioned several times in the text despite being excluded from the thematic analysis and results. I see that the PRISMA-ScR criteria encourages the inclusion of "gray literature" in the interest of creating a comprehensive review (checklist tip sheet Item 7). I think a more detailed rationale for exclusion of ISSW papers and a more explicit discussion in the Discussion and Limitations sections would help

readers understand the actual scope of this study, since most will be familiar with the ISSW and will wonder why it was not included as a source.

**Answer:** We realize that we have been unclear and have revised the description of how we have approached and motivated our decision for not including the ISSW papers in this paper.

As we explain, the reason for why we chose not to do so is that there is a very large spread in quality, and it is difficult to create stringent eligibility criteria (we chose to exclude PhD and MSc theses that have not been published peer-review for the same reason). However, given their importance in the field we still choose to search through the relevant databases, sort and extract data from the ISSW proceedings the same way we did with the peer-reviewed papers.

Section 3, Results: The authors' framing of each theme in the broader literature followed by their specific results is wonderful and will be exceedingly useful for future researchers. Very nicely done.

**Answer:** Thank you for this very encouraging comment!

1. Section 5, lines 532-3. The authors indicate that their study is a starting point for future work, and it certainly is. But a bit more detail would be helpful describing the specific future work the authors feel would be worthwhile (PRISMA-ScR checklist item 21). Of special interest would be what systematic reviews would be valuable and what specific aspects of "avalanche education quality" could be examined by future studies. The results of Section 3 provide a ready-made template for such a discussion, which I think would be valuable for many readers.

**Answer:** Thank you for this enlightening comment. We completely agree that this is lacking in the paper and have added a section in the discussion where we elaborate on the future work needed, both in terms of a more detailed content analysis of the existing research, and in terms of research on specific topics within the field of "human factors in avalanche terrain".

**Technical corrections**

1. Page 1, line 19: The abstract states 100 ISSW papers; the ISSW coding file contains 81. I think it would help readers to know up front that these papers were excluded from the paper's results.

**Answer:** Thank you for noticing this error of ours. This is explained by the criteria used for selecting the ISSW papers (described in the answer to comment 4). We have clarified this througout the text

Page 1, line 22: Begin sentence with Twelve not 12.

**Answer:** Thank you for noticing our sloppy use of numbers. We have change to "Twelve"

1. Page 2, line 61: reveal is misspelled.

**Answer:** thank you for making us aware of this typo.

1. Page 12, line 398: trend(s)

**Answer:** Thank you for noticing this. We will change from "trend" to "trends"

1. Page 12, line 423: decision-making aid(s)

**Answer:** Thank you. We will change to "decision-making aids"

1. Page 13, line 446: ...studies include...

**Answer:** Another sloppy mistake of ours. Thank you. We will change from "includes" to "include".